# Evolution and Interspecies Transmission of Canine Distemper Virus—An Outlook of the Diverse Evolutionary Landscapes of a Multi-Host Virus

**DOI:** 10.3390/v11070582

**Published:** 2019-06-26

**Authors:** July Duque-Valencia, Nicolás Sarute, Ximena A. Olarte-Castillo, Julián Ruíz-Sáenz

**Affiliations:** 1Grupo de Investigación en Ciencias Animales-GRICA, Facultad de Medicina Veterinaria y Zootecnia, Universidad Cooperativa de Colombia, sede Medellín 050012, Colombia; 2Sección Genética Evolutiva, Facultad de Ciencias, Universidad de la Republica, Montevideo 11200, Uruguay; 3Department of Microbiology and Immunology, UIC College of Medicine, Chicago, IL 60612, USA; 4Facultad de Ciencias Exactas, Naturales y Agropecuarias. Universidad de Santander (UDES), sede Bucaramanga 680002, Colombia

**Keywords:** genome evolution, canine distemper virus, hemagglutinin gene, genotype

## Abstract

Canine distemper virus (CDV) is a worldwide distributed virus which belongs to the genus *Morbillivirus* within the *Paramyxoviridae* family. CDV spreads through the lymphatic, epithelial, and nervous systems of domestic dogs and wildlife, in at least six orders and over 20 families of mammals. Due to the high morbidity and mortality rates and broad host range, understanding the epidemiology of CDV is not only important for its control in domestic animals, but also for the development of reliable wildlife conservation strategies. The present review aims to give an outlook of the multiple evolutionary landscapes and factors involved in the transmission of CDV by including epidemiological data from multiple species in urban, wild and peri-urban settings, not only in domestic animal populations but at the wildlife interface. It is clear that different epidemiological scenarios can lead to the presence of CDV in wildlife even in the absence of infection in domestic populations, highlighting the role of CDV in different domestic or wild species without clinical signs of disease mainly acting as reservoirs (peridomestic and mesocarnivores) that are often found in peridomestic habits triggering CDV epidemics. Another scenario is driven by mutations, which generate genetic variation on which random drift and natural selection can act, shaping the genetic structure of CDV populations leading to some fitness compensations between hosts and driving the evolution of specialist and generalist traits in CDV populations. In this scenario, the highly variable protein hemagglutinin (H) determines the cellular and host tropism by binding to signaling lymphocytic activation molecule (SLAM) and nectin-4 receptors of the host; however, the multiple evolutionary events that may have facilitated CDV adaptation to different hosts must be evaluated by complete genome sequencing. This review is focused on the study of CDV interspecies transmission by examining molecular and epidemiological reports based on sequences of the hemagglutinin gene and the growing body of studies of the complete genome; emphasizing the importance of long-term multidisciplinary research that tracks CDV in the presence or absence of clinical signs in wild species, and helping to implement strategies to mitigate the infection. Integrated research incorporating the experience of wildlife managers, behavioral and conservation biologists, veterinarians, virologists, and immunologists (among other scientific areas) and the inclusion of several wild and domestic species is essential for understanding the intricate epidemiological dynamics of CDV in its multiple host infections.

## 1. Introduction

Canine morbillivirus also known as canine distemper virus (CDV) belongs to the genus Morbillivirus within the family *Paramyxoviridae*, which also includes measles virus (measles morbillivirus—MeV), phocine distemper virus (phocine morbillivirus—PDV), peste des petits ruminants virus (small ruminant morbillivirus—PPRV), and rinderpest virus (rinderpest morbillivirus—RPV), which has already been eradicated [1,2]. CDV has the ability to infect a wide range of species and poses a risk to the conservation of free and captive wildlife [3,4]. The present review will analyze different viral and host characteristics to elucidate the conditions that have allowed the CDV to have this wide distribution, evidencing different successful genetic and evolutionary scenarios in both domestic and different wildlife environments.

The virus is pleomorphic (spherical and filamentous shapes), with a size between 150–300 nm in diameter and contains a genome comprising single-stranded, non-segmented ribonucleic acid (RNA) of negative coding sense. The genome comprises 15,690 nucleotides (15.6 kb), including six gene regions organized in separate and non-overlapping transcriptional units coding for six structural proteins: the nucleocapsid protein (N), phosphoprotein (P), viral polymerase (L), matrix (M) and the hemagglutinin (H) and fusion protein (F) glycoproteins [1].

The transcriptional CDV units are separated by untranslated regions (UTRs) that are relatively uniform (107–155 nucleotides), except for the UTR between the M and F genes, which has approximately 405 nucleotides. This UTR modulates virulence through the translational control of the protein F [5]. The *phosphoprotein* (*P*) gene (1524 nucleotides) is highly conserved and encodes a viral polymerase cofactor protein (P) and two additional nonstructural proteins V and C, which are produced by RNA editing and an alternative start of the translation, respectively [6]. The protein V suppresses the innate immune response by inhibiting the induction of type I and II interferons and the activity of the NF-kappa B complex in the host cell [7,8]. It has been established for MeV that the protein C suppresses the IFN-γ signaling pathway by inhibiting dimerization of the phosphorylated STAT1 protein [9,10]. The large protein (*L*) gene (6555 nucleotides) is highly conserved and encodes for the viral polymerase, that transcribes and replicates the virus RNA. The *N* gene (1572 nucleotides) encodes for the N protein, which encapsulates the viral RNA [1]. The P, L, and N proteins together make up the transcription/replication complex, known as ribonucleoprotein [6]

The *hemagglutinin* gene has 1824 nucleotides and encodes for the protein H, which shows the highest variability in CDV when compared to other morbilliviruses [11]. This gene has the highest divergence within CDV genome with substitution rates between 5.4 × 10^−4^–1.8 × 10^−3^ nucleotide substitutions per site, per year [11]. The protein H determines the viral tropism and initiates the infection by binding to the signaling lymphocytic activation molecule (SLAM) receptor in immune cells [12,13], and nectin-4 receptor in epithelial cells [14]. When dogs recover from natural infection/disease due to CDV, they develop neutralizing antibodies against the H protein which confers lifelong immunity to the diseases [15]. It has been shown that for this gene wild type and vaccine strains are highly divergent (7–10% for nucleotides and 8–11% for amino acids) with a nucleotide identity between 93–90% and amino acid identity between 92–89%. It is because of this high variability, that this gene is used in for the phylogenetic classification of CDV in different lineages or genotypes [16]. The fusion (F) protein is encoded by the *F* gene, composed by 1989 nucleotides. After the binding of the protein H to the host cellular receptor, the F protein mediates the fusion between the viral envelope and the host plasma membrane at physiological pH [17,18]. A short region of this gene (405 nucleotides) encodes for the F signal peptide; the signal peptide region is highly variable and has also been used in phylogenetic classification, representing a rapid method for characterizing circulating CDV strains [19]. The matrix (M) protein, encoded by the *M* gene with 1008 nucleotides, constitutes the inner layer of the viral envelope and interacts with the cytoplasmic domains of the H and F proteins [1]. When this protein is synthesized in the cytoplasm during the replication cycle, it also associates with the ribonucleoprotein complex, connecting it to the envelope, thus playing an essential role in viral shedding [18].

## 2. Disease in Domestic Dogs and Wildlife

Canine distemper is a worldwide distributed viral disease that is highly contagious and has high morbidity and mortality. CDV infection may result in subclinical (asymptomatic) or clinical disease manifestation. The main route of efficient infection is thought to be due to contact with fomites or via the aerosol route by nasal secretions [20]; however transmission through nasal contact with bodily excretions such as urine and feces is also likely [21].

For the development of the clinical disease, CDV infection has to go through two phases: first, there is an acute infection of the lymphatic system [22]; and second, there is an invasion of epithelial cells followed by viral shedding which allows transmission to other susceptible individuals [22]. CDV can also invade the central nervous system probably mediated by a putative third receptor called GliaR located in glial cells [23,24,25]. The clinical manifestations of CDV infection include depression, anorexia, nasal and ocular mucopurulent secretions, gastroenteritis, hyperkeratosis of the plantar pads and snout, and tooth enamel hypoplasia in puppies [26] In addition, seizures, myoclonia with hyperesthesia, ataxia, paresis, and tremors indicate central nervous system involvement [24].

CDV infects multiple species within the order Carnivora, including domestic dogs (*Canis lupus familiaris,* family *Canidae*), and various wild species within Canidae (hereafter referred to as canids) and other families such as *Felidae, Procyonidae, Mustelidae, Hyaenidae, Ursidae, and Viverridae* (hereafter referred to as non-canids). Some studies have also reported CDV infection in other mammalian families, such as *Cricetidae* (rodents), *Cercopithecidae* (Old World monkeys), *Suidae* (pigs), and *Elephantidae* (elephants) (for a comprehensive review, see [3]). Furthermore, recently CDV was reported as the causative agent of the neurological signs observed on a captive southern tamandua (*Tamandua tetradactyla,* family *Myrmecophagidae*, order Pilosa) in Brazil [27].

Viruses with a broad host range such as CDV could persist in populations with great sizes and may potentially cause disastrous epidemics in endangered species. Thus, the understanding of CDV’s epidemiology in multiple species is of great importance from a conservation point of view [28,29,30]. For example, different studies have reported the link between CDV infection and the population decline of various wildlife species [3,28], highlighting the possible role of CDV as a cause of extinction of certain endangered wildlife species [31]. CDV outbreaks have been reported in near-extinct species, such as the giant panda (*Ailuropoda melanoleuca*) in a wild animal rescue and conservation center in China with a morbidity of 27% and mortality of 23% [28] and the Ethiopian wolf *(Canis simensis)* with a reported mortality of up to 68% in a wild population in Ethiopia [29]. CDV has also been reported in monkey breeding sites with an incidence of 20–60% and a mortality of 5–30% in *Macaca fuscata* and *Macaca mulata* [32]. In the Serengeti National Park, in Tanzania, CDV has been detected in canid and non-canid species with varying outcomes of the disease depending on the genetic strain involved [33], with changes in the CDV genome of the non-canid viruses that were consistent with expansion of selected variants in non-canid hosts [34,35,36]. Most notably, a strain adapted to non-canids but not with the recent spillover of a dog strain caused a fatal epidemic in 1993–1994 in two non-canid species, killing approximately one third of the African lion population (*Panthera leo*) and several juveniles spotted hyena *(Crocuta crocuta)* [33].

In the United States, a mortality rate of 45% was reported in free-living raccoons (*Procyon lotor)* found within, around, and outside a zoo in 2001 [37]. CDV has also been stated as one of the most important causes of mortality in the gray fox (*Urocyon cinereoargenteus*) [38]. Likewise, a CDV outbreak almost led to the extinction of the black-footed ferret (*Mustela nigripes*) [39].

Epidemiological data from Europe show varying rates of CDV seroprevalence in wildlife species, particularly in red foxes (*Vulpes vulpes*) with a prevalence of 4% to 30.5% in countries such as Spain, Portugal, Italy, and Germany [30,40,41,42]. In Central and South America, authors have reported the presence of CDV in 2% of wild felids in Costa Rica [43], while in Brazil CDV seroprevalence has been reported in multiple free-ranging wild carnivore species with a prevalence ranging between 10.6 to 23% [44,45,46].

The lethal epidemics of CDV in multiple species, have increased the understanding of the capacity of this virus to cause high mortality in carnivores, impacting negatively wild populations [47]. Those data allow us to propose that CDV, by infecting endangered wildlife species, decreases the populations of these species, facilitating its extinction.

The role of the adaptive immunity and host immune response to the CDV, has been widely discussed and studied in the domestic dog [48]. However due to the impossibility to experimentally study this topic in wildlife, it is only possible to speculate how the evasion of the host immune system is involve in CDV infection in different species and its role in the pathogenesis.

## 3. Reservoirs vs Viral Adaptation: Two Different Transmission Sceneries

### 3.1. The Role of Reservoirs in CDV Epidemiology

CDV transmission between domestic dogs and wildlife depends on the dynamics between their populations [49], genetic characteristics of the virus [50], host receptors [51] and other factors that are not fully understood.

Although the initial spread of CDV resulting from the interactions between domestic canids and wildlife has been shown to trigger high mortality in wildlife [49], CDV spreading from wildlife to domestic canids is also possible [52,53]. In 2015, authors analyzed CDV outbreaks in dogs and lions in the Serengeti National Park, revealing that initially the outbreaks in lions were due to virus transmission by domestic dogs; however, over the years the outbreaks in lions (1993–1994, 1999 and 2007) were asynchronous with those in domestic dogs, suggesting that CDV may persist in other wild species interacting with this lion population [3,49] and possibly in other ecosystems. This hypothesis is supported by the variety of species reported as susceptible for CDV infection [3].

Molecular phylogeography analysis of the origin and global evolution of CDV found that the probability of CDV first infecting domestic canids and then wildlife is higher than that for CDV infecting wildlife at first. Analyzing 208 sequences of the *H* gene the study propose that CDV strains emerged in the United States during 1858–1913 and from there through migrations—perhaps pet trade-it expands to Europe, Asia, Africa and America. Thus, domestic canids are probably the main CDV reservoir worldwide [54].

Due to the lack of knowledge about the ecoepidemiology of the virus in wild-life ecosystems, it is necessary to look for virus reservoirs, defined as one or more epidemiologically connected populations or environments in which the CDV can be permanently maintained and from which infection is transmitted to the defined target population. Populations in a reservoir may be the same or a different species as the target population [55,56].

In the case of CDV in peridomestic settings, CDV seropositivity has been reported in clinically healthy animals belonging to various orders and families. Although it is not known if those animals showed clinical signs before the studies, these findings reveal that CDV virus circulation in wild ecosystems occurs through certain species acting as a “meta-reservoir”. Those wild animals transmit the virus to diverse interconnected populations, explaining the asynchronous outbreaks of CDV reported in wildlife ecosystems in relation to domestic dog outbreaks [3]. This complex situation has been reported for CDV in hyenas and cheetahs months after a CDV outbreak in lions of South Africa in 2015 [4].

A similar scenery was established in the United States when two strains with just 56 different nucleotides in their genome, and belonging to the same CDV lineage, were found circulating in different hosts of distant geographical areas [57]. However, in this case the raccoon (*Procyon lotor*) has already been identified as a second CDV reservoir given that the target population is wildlife and the primary reservoir is the domestic dog [53]. The raccoon populations are often found in peridomestic habits, having contact with infected dogs, thus spreading the infection to other wildlife in wild ecosystems and to healthy dogs in urban ecosystem [53,57].

### 3.2. Role of Asymptomatic Animals in CDV Transmission

Multiple epidemiological scenarios can lead to the presence of CDV in wildlife even in the absence of infection in domestic populations [3]. Although CDV has been linked to the mass death of wild species both free-living [58] and in captivity [59], CDV can also occur in different domestic or wild species without clinical signs. For example, domestic dogs can be CDV-positive without developing signs but can still transmit the virus since viral RNA is found on fecal swabs, thus these individuals act as potential CDV reservoirs [60,61]. Recently, CDV shedding was observed up to several months after the cessation of clinical signs in recovered domestic dogs [62]; therefore emphasizing the role of asymptomatic domestic dog carriers in CDV epidemiology in this species in urban areas.

The report of an outbreak in giant panda (*Ailuropoda melanoleuca*) suggests that subclinical disease is also possible in this species, as during the outbreak one individual did not show clinical signs despite the detection of CDV genome from blood and nasal swabs samples. This animal had been previously vaccinated suggesting that the protective immune responses elicited by vaccination against CDV were not sufficient to prevent CDV infection acquired naturally, but may have attenuated disease, as this was the only individual that survived the outbreak [28]. In other wild species such as raccoons (*Procyon lotor*) and gray foxes (*Urocyon cinereoargenteus*) a high percentage of asymptomatic individuals were positive for CDV (55%) [63]. These could be a source of the virus, especially in the coldest months, when the virus can survive longer outside the host [63].

On the other hand, the experimental inoculation of cats (*Felis silvestris catus*, family *Felidae*) and domestic pigs (*Sus scrofa domesticus*, family *Suidae*) with a CDV virulent strain for domestic dogs resulted in infection and viral replication in lymphoid tissues and macrophages, but without clinical signs or virus shedding [64]. Interestingly, the inoculation of two cats with a strain that was lethal in a captive Chinese leopard *(Panthera pardus orientalis,* family *Felidae*) resulted in viremia in peripheral blood mononuclear cells and mild transient leukopenia and lymphopenia, however no clinical signs of disease were observed in the domestic cats [59].

‘Silent’ CDV epidemics have been observed in the Serengeti–Mara ecosystem, where serological analyses have shown an increase in the exposure to CDV in spotted hyenas and lions without obvious clinical signs, high mortality, or a reduction in population size [49,65,66,67], as observed during the lethal epidemic of 1993–1994 in these species. Spotted hyenas from the Maasai Mara National Reserve were more exposed to CDV than those near to human settlements, showing that in this case, domestic dogs may not be the only animals that transmit CDV to these wild species [66].

These results show the importance of long-term research when studying CDV in free-ranging wild species in which virus detection and the recognition of clinical signs of disease (or their absence) is quite challenging [33,68,69]. For this reason integrated research incorporating the expertise of wildlife managers, behavioral and conservation biologists, veterinarians, virologists, and immunologists (among other scientific areas) and the inclusion of several wild and domestic species is needed for understanding the intricate epidemiological dynamics of CDV in multiple hosts. By monitoring CDV in target populations, regardless of whether clinical signs are observed or not, the complex scenarios of CDV transmission may be explored; including the possibility of the circulation of different strains that may vary in their virulence in different species [33], the epidemiological importance of asymptomatic individuals in the spread of the disease [61], the existence of dead-end hosts—which are asymptomatic and do not shed the virus—such as the cat and pig [70] or perhaps either lions or hyenas when infected with CDV canid strains [33].

Multidisciplinary research is also needed to implement accurate strategies to mitigate CDV infection, particularly at the interface between wildlife and domestic animals. For example, limiting the contact between wild and domestic species [71] or administering vaccines to domestic animals to reduce the impact of CDV on populations at risk of extinction [3] have been proposed for different populations. Although the vaccination of wild carnivore species has been proposed based on successful examples of vaccination against CDV in different species [28,39,72,73], there are also multiple cases where after vaccination, typical CDV clinical signs have appeared with fatal consequences in different wild carnivore species [38,74,75,76], including those at risk of extinction [28,71,77,78]

In these last cases it is not possible to differentiate between vaccine failure—a vaccine that does not provide protection against wild CDV strains [77,79]—or possible vaccine escape—the ability of the wild strains to re-infect or reactivate the infection in the presence of neutralizing antibodies that cannot neutralize the virus, caused by mutations that change surface antigens [80,81]. However, as the effectiveness of different vaccines (adapted modified live vaccine in canid cells or in eggs) or even including the canary pox vectored vaccines can vary even between closely related species [38,74,82], the efficacy, safety, and possible risks of administering vaccines to free-living wildlife should be critically evaluated to avoid possible lethal consequences, especially in areas of high biodiversity. Moreover, the handling of the animals to administer the vaccine [83,84] or the vaccine itself [85] may result in negative immunological consequences for the animals, leaving them susceptible to CDV and other pathogens.

### 3.3. Viral Adaptation: the Role of Specialist vs Generalist Strains

The study of the factors determining viral host range is critical for our understanding of the diversity, evolution, and emergence of CDV and for the potential to predict virus changes in a given host [86,87,88,89]. One of the dominant forces driving virus evolution is mutation, which generate genetic variation on which random drift and natural selection can act, shaping the genetic structure of virus populations [86,90].

Concepts such as fitness—defined as ”the capacity of a virus to produce infectious progeny in a given environment” [90], quasispecies—“collections of closely related viral genomes subjected to a continuous process of genetic variation, competition and selection” [91], coevolution—“evolutionary change in a trait of the individuals in one population in response to a trait of the individuals of a second population, followed by an evolutionary response by the second population to the change in the first” [92], and antagonistic pleiotropy—defined as ”a beneficial mutation in one environment is either harmful or neutral in another environment, or mutations that are neutral in the environment in which they arose are deleterious in another” [93]—have been suggested to play a role on the adaptation and evolution of CDV in its different hosts.

For example, CDV is thought to have specialized in two major host orders: Caniformia and Feliformia, in which the virus quasispecies with better host fitness are considered specialized strains [33,93]. Those have acquired the ability to interact with SLAM receptors of different species, and they have coevolved with their hosts, thus fixing mutations allowing the virus to specialize in a particular host; thus, antagonistic pleiotropy in CDV shows that substitutions at the hemagglutinin are related to species jump. However, the capacity for evasion of the immune response should be studied in the other viral proteins in each affected host [93,94].

Virus adaptability to different environments can be viewed in terms of an adaptive landscape in which the peaks and valleys represent fitness estimates of the relative ability of the virus to enter and successfully replicate in an environment, such as different host species [90]. Functional mutations allow the virus to move through the fitness landscape, and selection helps fix adaptive mutations [95,96]. CDV cross-species transmission can be viewed within this adaptive landscape (Figure 1) in which each peak represents the optimized viral fitness of a virus strain in a given host. These peaks are separated by a fitness gradient because mutations optimizing the ability of a viral strain to successfully infect a different host can reduce its fitness in the original host, a phenomenon known as antagonistic pleiotropy [97]. In this evolutionary scenario, viral genetic variation is studied in a coevolutionary framework to understand the degree to which hosts influence virus diversity.

A comprehensive comparative genetic study using several complete CDV genomes showed that all the residues found to be under positive selection in CDV are located within the H protein, which is the viral protein interacting directly with the SLAM receptor in immune cells [50]. Within the receptor binding region of the H protein, substitutions for one of these residues (site 549) were systematically observed in non-canid hosts, pointing to their importance in host adaptation (Figure 1b) [50,94]. It is hypothesized that even when a few substitutions are needed for successful cross-host infection, the likelihood of extinction of ‘partially-adapted’ viruses (i.e., those in the fitness valleys) should be high because their fitness is not optimal in either host and therefore would be out-competed by fitter strains (i.e., those at the fitness peaks) [87]. However, for CDV the results of fitness analyses to assess the impact of substitutions at site 549 for cell entry and viral replication in canid and non-canid hosts revealed an alternative scenario for the reconstruction of viral evolution when considering fitness trade-offs across hosts: the evolution of specialist and generalist traits [93,94].

In a homogeneous environment with a single host species, strong coevolution promotes the optimization of viral fitness in that host (i.e., fitness peaks), resulting in low fitness of the virus in other species (i.e., fitness valleys). Therefore, specialists should be favored in homogeneous environments [97]. By contrast, in heterogeneous environments where multiple hosts are available, the evolution of generalists with the highest average fitness that allow the virus to infect all hosts should be favored, which results in suboptimal fitness in any single host in comparison to single-species specialists [98]. Therefore, specialists should have greater fitness in the host species to which they are well adapted than generalists in that same host [99].

For the case of CDV, the large domestic dog world population could represent such a favorable, homogeneous environment that permitted the optimization of CDV fitness in this particular host. Accordingly, fitness assays showed that the CDV-H protein from domestic dog had a higher performance in cell lines expressing domestic dog than non-canid (African lion and domestic cat) SLAM, as expected in antagonistic pleiotropy. On the other hand, non-canid CDV-H proteins performed equally well regardless of which host SLAM (canid or non-canid) was expressed, but the overall performance was lower than that of domestic dog CDV-H protein in cells expressing domestic dog SLAM. Moreover, the substitution of site 549 from a canid to non-canid residue in the domestic dog CDV-H protein showed a reduction in fitness in domestic dog cells [93,94]. Together the results of the comparative genetic analysis and the fitness assays of CDV in canid and non-canid species allow another view of the model of virus cross-species transmission (Figure 1) that is consistent with the presence of specialist (i.e., fitness peak in the domestic dog) and generalist (i.e., shallow fitness valley, possibility to infect several species) traits (Figure 1) in the relation of the CDV-H protein with the SLAM receptor of canid and non-canid hosts.

The diverse examples shown above show that the transmission and epidemiology of CDV may be driven by three different mechanisms: (1) dissemination by means of animals with subclinical infection which can transmit virus; (2) in urban and peridomestic settings where wild animals come into direct contact with domestic animals which in this case would be considered reservoirs, and (3) strains that have adapted to a specific host mainly driven by the existence of specialist vs. generalist strains.

## 4. CDV and Measles: Complementary Infection Models

Both CDV and MeV belong to the genus *Morbillivirus* and have the same genomic architecture. The high nucleotide and amino acid similarities between both viruses results in high levels of functional and structural conservation, this is why the CDV has been used as a model to understand the pathogenesis of measles, and often has been used in genetic engineering to develop vaccines and antiviral treatments [100,101].

Previous studies have shown that morbilliviruses including CDV, phocine distemper, rinderpest and peste des petits ruminants virus circulate in nature [102] by using two primary receptors: SLAM, expressed in cells of the innate immune response, and in activated T and B cells [51], and nectin-4, expressed in polarized epithelial cells [103]. The SLAM receptor is considered the main morbillivirus receptor because it allows its initial entry into cells of the immune system [104] Comparative genetic analysis between SLAM sequences from different species—domestic dog (*Canis lupus familiaris*) and domestic cat (*Felis catus*) [105]—revealed four important interaction sites with MeV and CDV including amino acids at positions 123, 127, 129, and 131 [102,105,106].

The nectin-4 receptor is involved in cell-to-cell propagation of CDV and MeV, and also in cell-cell fusion (syncytia formation) [103,107]. Four binding sites on the nectin-4 receptor have been identified (F132, P133, A134, and G135) for the optimal binding to CDV H both in dogs and humans [107]. In both species, nectin-4 is expressed in epithelial cells [103,108], however it has been found that in domestic dogs nectin-4 is also found in neurons, ependymal cells, epithelia of choroid plexus, meningeal cells, granular cells, and Purkinje’s cells, which could indicate its partial involvement in the neurovirulence of CDV observed in dogs [109]. Besides the presence of nectin-4 in the white matter or astrocyte cultures of dogs has not been confirmed, suggesting the existence of a third cellular receptor [109].

Indeed, it has been hypothesized the presence of a viral receptor, found mainly in astrocytes, which allows the noncytolytic cell-to-cell transfer of CDV and its spread into the white matter. This proposed third receptor was named glia R [23], which other authors have reported as the means of colonization of other morbilliviruses in the brain of dolphins [110]. However, all authors conclude that this new receptor should be fully identified and characterized in canines because it is very important for the neurovirulence of CDV and other morbilliviruses [23,109,110].

The SLAM receptor has 39 residues that are involved in the interaction with CDV protein H [105]; when comparing the aminoacidic sequences of the SLAM receptor in different species -human, dog, raccoon, mink, fox, cow, mouse- an identity of 57.2% is observed between the human SLAM and the dog, fox and cow receptors. Likewise, there are an amino acid identity of 93.1% between nectin-4 receptor in humans and dogs [101]. The possible explanation for the inability of CDV to infect humans and domestic cats could be the differences in certain residues that alter the SLAM interaction region determining a low affinity interaction between the cellular receptor and CDV protein H [105].

Naturally acquired and experimental inoculation of CDV in non-human primates have shown high neuropathogenicity potential in those animals with severe encephalitis [32,111,112,113]. Thus, the ability of the virus to adapt to human cells and to infect humans is subjected to speculation [3,114]. It has been established that CDV can gain the ability to infect human cells by acquiring certain specific mutations in the protein H [106,115,116]. Vero cells expressing human SLAM were infected with CDV and in the third passage syncytia formation was observed; when analyzing substitutions of the protein H a single amino acid change was found (Table 1) which allowed the CDV-H to use the human SLAM receptor [106]. On the other hand, the human cell line H358 (expressing Nectin-4 receptor) was infected with CDV and authors find mutations in H and other proteins at the eighth passage [116] (Table 1). In another study by Otsuki et al. [115], the authors showed that CDV was adapted to the two human receptors in H358 and Vero-human-SLAM cells by acquiring amino acid substitutions in the V/C, F and H proteins after 8 culture passages [115]. Taking together, these results allow to presume that, at least at the receptor level, humans are susceptible to CDV infection, however further studies must be done in order to establish if human cells could be also permissive to CDV replication [106,115].

Given the identity (86%) between the human and non-human primates SLAM receptor [13] and the similar pathogenesis between CDV and MeV, CDV infected animals have been used as study models for MeV replication in vivo. However, this similar pathology opens the question if CDV can cause the same pathogenesis in humans [117]. In the mid-1990s, an attempt was made to establish a possible association between CDV with multiple sclerosis by finding CDV antibodies in patients with multiple sclerosis [118].

It is well known that experimental MeV vaccination in pups usually leads to an immune response with cross-reactive activity to CDV even in the presence of maternal antibodies [119,120,121,122]. The produced antibodies recognize epitopes conserved between MeV and CDV thereby conferring a certain level of protection [123,124]. Therefore, it is possible to hypothesize that immunity against MeV may have protected humans against CDV infection or at least avoided the presentation of clinical signs [116]. In non-human primate models infected with CDV and vaccinated against MeV, the antibodies generated in these animals partially cross-react against CDV protecting them from the clinical disease [102,125]. The possibility that MeV vaccination stops in the future in several geographical regions due to the success of the eradication program, allows us to think that the CDV can possibly jump the species barrier, infect humans and cause clinical disease in humans similar to that already reported in non-human primates [106,115,125].

### Genetic Variability of CDV

As mentioned earlier, CDV protein H determines the cellular tropism and host range when interacting with the SLAM receptor in host lymphoid tissues [13] and with nectin-4 in epithelial tissues [103]. Therefore, the *H* gene has the highest variability and thus it has been widely used in genetic and phylogenetic analyses [126,127].

Studies using the *H* gene show a worldwide distribution of CDV genotypes, classifying the circulating strains into 17 lineages (Figure 2): America-1 (vaccine strains), America-2, America-3, America-4, America-5 Arctic-like, Rockborn-like, Asia-1, Asia-2, Asia-3, Asia-4, Africa-1, Africa-2, European Wildlife, Europe/South America-1, South America-2 and South America-3 [33,57,59,80,81,128,129,130,131,132,133].

These lineages are defined according to the amino acid divergence of the H protein between the different strains. Thus, it is considered that two given strains belong to a certain lineage when the protein H amino acid divergence is less than 3.5% [134]. On the other hand, following the guidelines based on MeV studies, sub-lineage strains present a minimum “bootstrap” value of 70% and at least 98% of aminoacidic identity [61].

Evolutionary comparative analyses of both virus receptor-binding proteins and host receptor proteins may help elucidate virus–host co-evolutionary dynamics. Additionally, comparative genomics in a phylogenetic framework permits the assessment of evolutionary forces that influence genetic variation in those interacting regions reflecting virus–host arms races [89,96]. Comparative genetic studies including several complete genome sequences of CDV showed that certain residues are under positive selection, almost all located in the receptor binding region of the H protein [50,93]. In particular specific substitutions in sites 530 and 549 were systematically observed in non-dog species pointing their importance in host adaptation [50].

Domestic dogs have eight possible substitutions at position 530 (G/D/N/E/R/S/A/K) and two substitutions at 549 (H/Y) present in at least ten lineages; in non-canid wild animals there are seven substitutions at position 530 (G/D/N/E/R/C/V) and two substitutions at position 549 (H/Y) present in seven lineages. On the other hand, wild canids have three substitutions at position 530 (G/N/R) and two substitutions at position 549 (H/Y) that are present in five lineages. (Table 2) [135].

Recent analyses of the CDV strain responsible for the outbreak in lions of Serengeti National Park in the 1990s demonstrated that all the species affected (including dogs and wild canids) presented the 530D substitution [33], agreeing with Liao et al. [135] who reported the 530 site is not directly related to host adaptation because it is conserved in different species. In addition, although an association has not been statistically established, it has been shown that dogs have a greater tendency to present 549Y, while substitution 549H occurs most commonly in wildlife, which shows a possible association between this substitution and the affected species [94].

Likewise, studies conducted in China characterizing different CDV isolates in mink breeding sites (non-canid wild species), raccoon dogs and foxes (wild canid species) showed the trend 530G and 549H [137]. In monkeys (non-canid species), there was a trend of 530G and 549Y with an additional substitution at another site of the protein H (542F) that may play an important role in CDV adaptation to primates (Table 2) [137].

However, in an outbreak in urban raccoons (non-canid species) in Europe, researchers found the trend 530G and 549Y, and concluded that the Y549H substitution apparently is not essential to facilitate infection in wildlife other than canids. The explanation for this outbreak was that urban raccoons are in contact with foxes (wild canids), which are widespread species in urban areas thus having great contact with domestic dogs. Therefore, foxes act as a wild reservoir of domestic dog strains in which the substitution 549Y is predominant [136].

Contributing to this debate, there was an 11.5% presence of the Y549H substitution in dogs from Brazil. In addition, all the dogs presented 530G and 580R, the latter representing 89% of the circulating strains (Table 2). The authors suggest that this trend may be associated with the condition of the “stray” dogs that were sampled, who are more likely to have contact with wildlife acting like reservoirs in these ecosystems [126].

Regarding to wildlife, more sequences of the *H* gene are needed to determine the trend of the presence of substitutions at positions 530 and 549 because the 549H substitution has not been reported in the lineages of Asia, South America, and Africa due to the lack of studies in wild species at these areas [94].

With respect to CDV jump to humans, the literature reports that substitutions D540G and M548T in the protein H, besides Y267C in the protein V, C116Y in the protein F, and M267V in the phosphoprotein have allowed cell invasion in H358 cells (Table 2). However, different authors suggest there must be mutations in other genes allowing the intracellular adaptation of CDV [106,115]. We point out that CDV complete genome sequences must be evaluated to identify other substitutions that can define its ability to jump between hosts, because most studies aimed to explain an adaptation process from a single component (protein H) ignoring the genetic changes in other regions that may contribute to the inter-species jump and adaptation.

## 5. Evolutionary Rates and Molecular Clock of CDV

Different studies have tried to elucidate the time of the most recent common ancestor (tMRCA) for CDV [11,54,126,139]. However, the number of sequences available in the GenBank throughout years has led to diverse results, making the data from different studies not comparable. A first approach, published in 2008, estimated the tMRCA for CDV to happened 58 years ago using a sequence from 2001 as reference, suggesting that the ancestor existed in 1943, with an interval from 1894 to 1974 and 95% higher probability density (HPD) [11]. Next, a tMRCA was estimated at 125 years by analyzing a strain from 2011, corresponding to the year 1886 with an interval from 1858 to 1913 and a 95% HPD [54]. In Taiwan, the tMRCA was established in 1945, with an interval from 1918 to 1966 and a 95% HPD [139]. Likewise, in Brazil, another tMRCA was found to be 92 years from 1919, with an interval from 1899 to 1944 and a 95% HPD [126].

The differences found in the tMRCA studies for CDV are ascribed both to the difference in the number of sequences evaluated in the different studies (from 35 to 208 sequences of the *H* gene) and to the geographical and temporal diversity of those sequences. In this sense, it is important to note that the intervals intersect and that more reliable results will be obtained by including sequences with greater temporal and spatial divergence in the analysis. A time-measured Bayesian maximum clade credibility (MCC) tree for the CDV *H* gene is presented in the Figure 3 by using 69 CDV H sequences retrieved from Genbank.

CDV evolutionary rates (substitution rates per site per year [subs/site/yr]) for the *H* gene were estimated in 4.8E-4 subs/site/yr, with 95% HPD and a range of 3.9 × 10^−4^ to 5.9 × 10^−4^ subs/site/yr, representing a high evolutionary capacity [54]. Other substitution rates have been reported, including 11.65 × 10^−4^ and 7.41 × 10^−4^ subs/site/yr [11,139]. This shows that substitution rates of CDV are slightly higher compared to other morbilliviruses such as MeV (6.5 × 10^−4^ subs/site/yr) [11]. The high genetic variability of CDV may have facilitated the inter-species jump, and its adaptation and specialization to other hosts [93]. It is important to remember that tMRCA studies for CDV and for other viruses are based on available sequences in the Genbank, generating a bias that must be taken into account in the analysis of the results, since it can not necessarily coincide with the historical records present in the literature as has been recently discussed by others [141].

## 6. Pathogenicity Predictions

Genetic approaches are one of the multiple tools used to analyze viral spread and pathogenesis. Association predictions between morbidity and lethality related to the presence of specific substitutions at certain genomic positions have been presented by several authors [33,126,139]. It is important to note that these predictions are hypotheses that arise from the analysis of the sequences obtained by the authors and that due to the lack large epidemiological studies no statistical association has been established.

In the non-canid CDV strains from the 1993–1994 Serengeti outbreak a new substitution in the H protein (R519I) was associated with the presence of clinical signs in these wild species (African lion and spotted hyena). In this study, the authors evaluated the presence of various combinations of substitutions at positions 519 and 549 of the protein H in strains from domestic dogs, wild canids, and non-canids wild species. They found that a rare combination of substitutions in these positions (519I and 549H) was only observed in non-canid strains and was related to a fatal outcome only in non-canids (lion and hyena). The other combinations found (519R/549Y and 519R/549H) caused death only in domestic and wild canids (Table 2) [33].

Similarly, in a Brazilian study all the strains isolated from dogs harbored the substitution 549H causing the death of every animal, with a statistically significant association (*p* < 0.05). In addition, the authors concluded that this substitution is associated with the creole breeds (*p* < 0.05) [126]. In dogs from Medellin, Colombia, our research group reported the presence of CDV of the South America−3 lineage and the combination of 549Y, 519R, and 530N; one dog presented 549H (Table 2). All of these cases showed evidence of nervous symptomatology and respiratory disease with fatal outcomes [129]. The mutations association studies of the *H* gene are the first performed for CDV, and should be extended to the study of the complete genome to explain the pathogenesis in the various susceptible species. It has been reported that mutations in intergenic regions, and in the 5′ and 3′ UTRs can affect the virulence and pathogenicity of field strains [142]. Specifically, mutations in CDV *M–F* intergenic region were associated with a different viral phenotype [5], highlighting the importance of conducting association studies with complete genomes.

## 7. *H* Gene vs. Complete Genome

There is controversy about which position substitutions in the *H* gene are associated with CDV host adaptation and pathogenicity in the various affected species, which has led the scientific community to conclude that more sequences of the *H* gene in wildlife should be obtained to clarify this point [94]. As was mentioned, it is necessary to broaden the search for molecular markers to explain relevant aspects such as the species barrier jump and the pathogenicity of field strains. Studies on CDV infection in human cells have characterized the complete genome of these adapted strains, and in addition to the *H* gene sites, substitutions associated with host adaptation and immune system evasion were found in other genes [115]. Additionally, other authors have characterized the complete genome of wildlife strains and have reported important substitutions (at CDV-H 519/549) that could explain the adaptation, immune system evasion, and the presence/absence of clinical signs in these animals [33].

Also, studies of CDV complete genome have reported homologous recombination between the P, H, and L genes in vaccine and field strains from different hosts [143]. In addition, phylogenetic studies from the complete genome reflect a more accurate strain classification [144]. Likewise, all glycosylation sites that may mask antigenic epitopes—which could explain vaccine failures—can be established as other sites of positive, neutral, or negative selection in CDV membrane proteins [57,143,145].

## 8. Conclusions

Viruses with a broad host range such as CDV are highly contagious and have high morbidity and mortality in wild and domestic animal populations. Epidemiological data from around the world show that CDV has become an important threat for conservation of multiple species even outside of the Carnivora order. The lack of ecoepidemiological information of CDV transmission beyond the infection between dogs, has led to investigate the importance of the infection in a multihost scenario that is not totally yet understood. The capability of the virus to adapt to multiple species and evolve in those, makes CDV not only a well-known pathogen of domestic dogs, but one of the most important pathogen of mammals with the ability to jump the species barrier and perhaps even infect humans. The different transmission scenarios discussed in this paper are an attempt to elucidate the transmission of CDV in different environments, and encourage uniting multiple areas of research for understanding the intricate epidemiological dynamics of CDV in multiple hosts.

The *H* gene is the basis for the phylogenetic classification of CDV due to the high evolutionary pressure to which its protein subjected. The *H* gene study has also allowed researchers to infer key amino acid substitutions involved in the inter-species jump. The developing of new studies of the CDV complete genome would enable us to explain in detail its evolution and to establish substitution rates, glycosylations, and homologous recombination points that would explain the pathogenicity, species jump ability and vaccine failure of this virus.

## Figures and Tables

**Figure 1 viruses-11-00582-f001:**
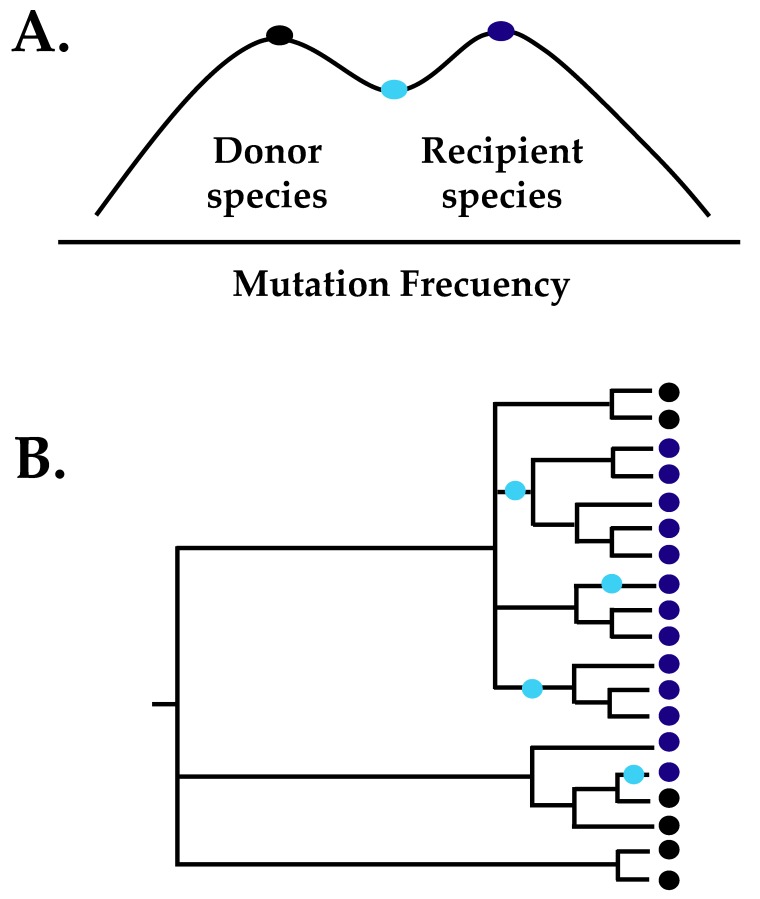
Evolutionary model for cross-species transmission of canine distemper virus (CDV). In the model, each peak represents the optimized viral fitness in a given host species, and specific mutations (colored circles) are essential to reach this optimum in the new host. (**A**) The two fitness peaks of CDV are separated by a shallow fitness valley, thus few adaptive mutations are required. One mutation allows the infection of other species but also decreases the fitness in the donor species (light blue circle, generalist trait). Coevolution with the new species can select for further mutations that optimize the fitness (dark blue circle, specialist trait). (**B**) CDV infection in canids and non-canids as an example of the model presented in (**A**). A simplified phylogeny of the *H* gene showing the relationships between CDV strains from canid and non-canid hosts, indicated at the tip of the tree as black and blue squares, respectively. The distribution of the amino acid site 549, which has been shown to be involved in CDV infection in non-canid hosts and is under positive selection, is mapped onto the branches of the tree. Commensurate with the coloring in panel (**A**), the substitution of site 549 is shown in light blue because it is considered a generalist trait that permits the infection of various non-canid species, but with lower fitness than the specialist trait in canids, marked in black.

**Figure 2 viruses-11-00582-f002:**
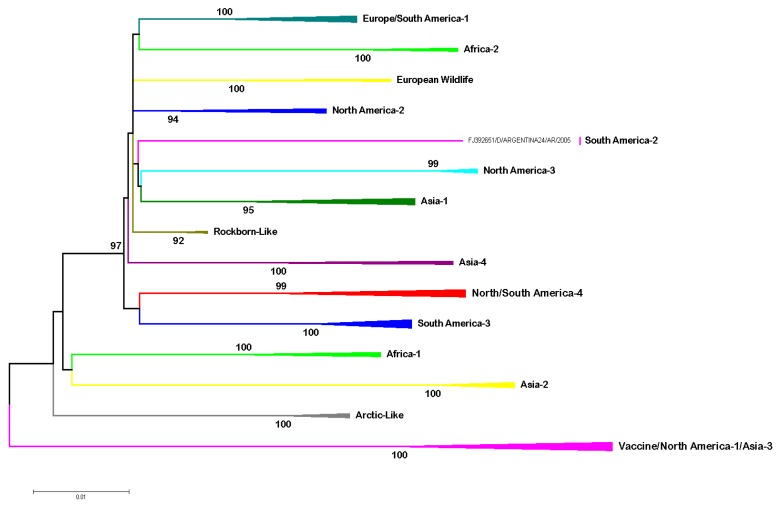
Phylogenetic relationship between 69 CDV strains based on the *H* gene sequence. The phylogenetic tree was obtained by the Maximum likelihood method with a Bootstrap of 1000 based on the evolution model T92 + G, 69 sequences were analyzed with a total of 1824 nucleotides. Evolutionary analyses were conducted in MEGA7.

**Figure 3 viruses-11-00582-f003:**
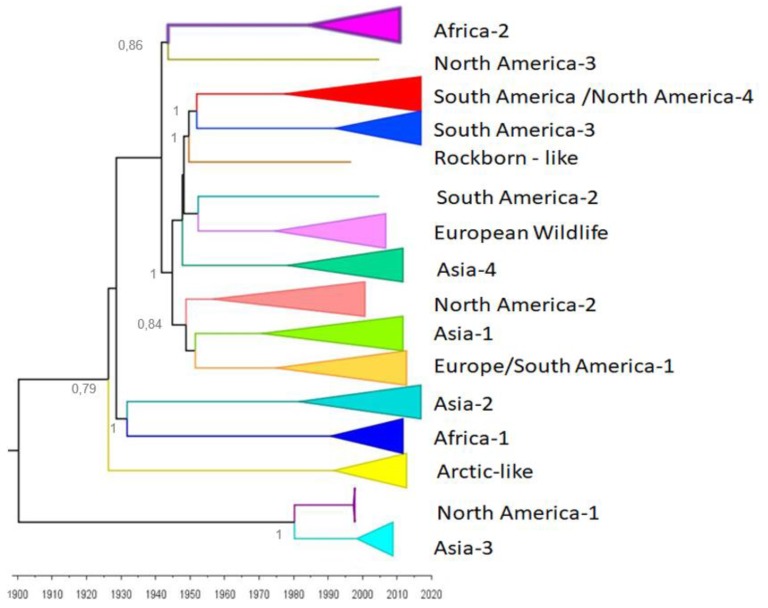
Time-measured Bayesian maximum clade credibility (MCC) tree for the CDV *H* gene using BEAST [140]. The time of the most recent common ancestor (tMRCA) of CDV is shown using 68 available sequences of the *H* gene for the lineages reported to date. Posterior probability limit: 0.7 are showed on nodes.

**Table 1 viruses-11-00582-t001:** Amino acid substitutions in the CDV *H* protein reported in strains adapted to human cell lines.

CDV				Author
*V* Gene	*F* Gene	*P* Gene	*H* Gene	
Y267C	C116Y	M267V	M548T	Otsuki et al. [115]
			D540G	Bieringer et al. [106]

**Table 2 viruses-11-00582-t002:** Summary of amino acids present at different positions of the H protein interacting with the SLAM receptor.

***H* Gene**					
**530**	**549**	**Animal**		**Lineage**	**Author**
G	Y	Dogs		Europe, Asia 1, America 2, Arctic-like, Asia 2	Nikolin et al. [94], Liao et al. [135]
G	Y	Wild canids		Europe, Asia 1,	Nikolin et al. [94], Liao et al. [135]
G	Y	Wild species (other than canids)		Europe, Asia 1,	Nikolin et al. [93], Rentería-Solís et al. [136], Liao et al. [135]
G	H	Wild canids		Europe, Asia 1,	Nikolin et al. [93], Panzera et al. [54], Liao et al. [135]
G	H	Wild species (other than canids)		Europe Asia 1, America 2	Nikolin et al. [93], Panzera et al. [54], Liao et al. [135]
G	H	Dogs		America 2, Europe, South America 1	Nikolin et al. [93], Liao et al. [135], Fischer et al. [126]
D	Y	Dogs		European Wildlife, South America 2	Nikolin et al. [93], Liao et al. [135]
D	Y	Wild species (other than canids)		European Wildlife, Africa 2	Nikolin et al. [33]
D	Y	Rockborn-like, Candur, China, Sweden		Vaccine	Nikolin et al. [93], Liao et al. [135]
N	H	Wild canids		European Wildlife	Nikolin et al. [93]
N	H	Dogs		European Wildlife, America 1	Liao et al. [135]
N	H	Wild species (other than canids)		America 1	Rentería-Solís et al. [136], Liao et al. [135]
D	H	Wild species (other than canids)		European Wildlife, Africa 2	Nikolin et al. [93], Rentería-Solís et al. [136], Liao et al. [135]
C	H	Wild species (other than canids)		European Wildlife	Nikolin et al. [93],
N	Y	Dogs		Arctic-like, Africa 1, America 1	Nikolin et al. [93], Liao et al. [135]
N	Y	Wild species (other than canids)		America 1	Nikolin et al. [93], Liao et al. [135]
N	Y	Wild canids		Arctic-like	Liao et al. [135]
E	Y	Dogs		Asia 2	Nikolin et al. [93], Liao et al. [135]
E	H	Wild species (other than canids)		Asia 2	Rentería-Solís et al. [136]
R	Y	Wild canids		Asia 2	Nikolin et al. [93]
R	Y	Dogs		Asia 2	Liao et al. [135]
R	H	Wild species (other than canids)		America 2	Nikolin et al. [93], Liao et al. [135]
R	H	Wild species (other than canids)		America 2	Rentería-Solís et al. [136],
S	H			Vaccines	Nikolin et al. [93], Liao et al. [135]
S	L			Vaccines	Liao et al. [135]
S	Y	Dogs		Arctic-like	Liao et al. [135]
A	Y	Dogs		Asia 1	Liao et al. [135]
V	H	Wild species (other than canids)		European Wildlife	Nikolin et al. [93], Liao et al. [135]
K	Y	Dogs		Asia 2	Liao et al. [135]
**Other Positions Reported in the *H* Gene**					
**542**	549	**Animal**		**Lineage**	**Author**
**F**	Y	Wild Species (Other Than Canids)		Asia 1	Zhao et al. [137]
			**Animal**	**Lineage**	**Author**
**530**	**549**	**580**			
**G**	H	Q	Dogs	South America 1	Fischer et al. [126]
**276**	**392**	**542**			
**V**	R	F	Wild species (other than canids)	Asia 1	Feng et al. [138]
**530**	**519**	**549**			
**N**	R	Y	Dogs	South America 3	Espinal et al. [129]
**Other Positions Reported in the *H* and *V* Genes**					
**GENE *H***	**GENE *V***		**Animal**	**Lineage**	**Author**
**519**	**149**	**134**			
**I**	H	S	Wild species (other than canids)	Africa 2	Nikolin et al. [33]
**R**	Y	G	Wild canids	Africa 2	
**R**	H	G	Wild canids	Africa 2	
**R**	Y	G	Dogs	Africa 2	
**I**	H		Wild species (other than canids)	America 2	

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
