# Peer review of "Evolution and Interspecies Transmission of Canine Distemper Virus—An Outlook of the Diverse Evolutionary Landscapes of a Multi-Host Virus"

_viruses, 2019, doi:10.3390/v11070582_

Reviewer 1 Report

This review by Duque-Valencia et al examines the evolution and interspecies transmission of canine distemper virus. In general, an attempt to place the existing CDV literature within a more rigorous evolutionary framework is to be welcomed,  and helps to differentiate this article from a number of  comprehensive well written reviews on CDV which have been published in the past 3-4 years. The main weakness to this approach is the paucity of high quality data, which is available to review, especially with respect to mechanistic in vitro or in vivo studies or large-scale whole genome sequencing of wildlife samples. In general the study is well written but is often repetitive with major flaws in the use of references and presentation of data and more concise writing would also be appreciated.

 Major comments

 1. Please completely reassess the use of references. There are many, many examples of references being used in appropriately in that they often appear to bear no relation to what is being discussed in the preceding sentence. By line 111, reference 1 which is itself a review (book chapter) has already been cited 12 times.  Just to state a couple of the most egregious examples: Line 353, reference 93; Lines 381-382, reference 20; line 412, reference 90.

2. The introduction needs to be modifies as it currently reads like part of a PhD thesis i.e. before describing the properties of CDV, I would suggest to start with a small paragraph introducing the overall topic of the review. Then I would group the description of the proteins involved with the RNP and the M/F/H into two separate paragraphs instead of one paragraph per gene.

3. For the section ‘Reservoirs vs Viral Adaptation: two different transmission sceneries’ (line 151), I would suggest that the authors start with the role of reservoirs in CDV epidemiology as the literature supporting this is much more extensive than the more speculative role of asymptomatic animals in CDV transmission. 

4. For the section ‘CDV and Measles’, I would consider having a new title which is more self-explanatory and starting with a small introductory paragraph which introduces this subject.

5.  It is not clear what the difference is between Tables 2 and 3 as some of the H amino acids listed in Table 3 also interact with CD150. I would suggest combining the references and alter the formatting so that the reference is given as a number to save on space.

6. I would consider deleting figure 3 as one of the weaknesses of including original data in a review is that the way the analysis was performed is not described, with no data for the reliability of the different branches or on what sequences were used for the analysis. In addition, the position of Asia 3 looks to be incorrect and the dates seem to bear no relation to the historical literature.

  Minor comments

 1. The title is too long and unwieldy, why not keep it to ‘Evolution and interspecies transmission of canine distemper virus’.

2.  Please check the formatting of the references.

3.  There are many English mistakes throughout the manuscript. Please do additional proofreading.

4.  Lines 71-72. The functions of all of the proteins has been described except the viral polymerase – maybe add that it replicates and transcribes the viral genome.

5. Line 132. There is currently no evidence to suggest that the CDV strain responsible for the outbreak of distemper in lions in 1993/94 was adapted to non-canids. Please change.

6.  Lines 147-148. This is a repetition of lines 134-136.

7.  Lines 230-232. How does the proposal of CDV spread from USA to Europe during 1858 to 1913 match up with the extensive documentation of CDV outbreaks in Europe from 1760 e.g. see the famous description of CDV by Edward Jenner in 1809 (20895116).

8. Lines 271-271. All of the available evidence shows that CDV has specialized in the order Caniformia, with infections of large felids largely representative of spill over events from infected canid species.

9.  Lines 366-371. This is written as if a third CDV receptor has already been discovered. Before ‘this new receptor should be further characterized’, it first needs to be identified. It should also be mentioned that in CDV-infected brains, neurons are more commonly infected than glial cells.

10. Line 412. This reviewer is unaware of any countries in which MV vaccination has been discontinued due to successful eradication. Indeed, it is worth noting that there is yet no official eradication program, although on a regional level e.g. the Americas, endemic MV transmission has been stopped.

11. Figure 2. Please re-check the position of the Asia 3 lineage as this does look to be correct. 

Author Response

RESPONSE TO REVIEWERS COMMENTS (Manuscript ID: viruses-482333)

Evolution and interspecies transmission of Canine Distemper Virus – an outlook of the diverse evolutionary landscapes of a multi-host virus

July Duque-Valencia, Nicolás Sarute, Ximena A. Olarte-Castillo, Julián Ruíz-Sáenz*

Dear Editor

Please find below our point by point responses to the comments regarding our Manuscript ID: viruses-482333, formerly entitled “Evolution and interspecies transmission of Canine Distemper Virus – an outlook of the diverse evolutionary landscapes of a multi-host virus”. The changes are highlighted in Yellow in the file.

We would like to thank the Reviewer for their helpful suggestions, for critical analysis of the manuscript, and for providing new discussion topics.

REVIEWER 1

This review by Duque-Valencia et al examines the evolution and interspecies transmission of canine distemper virus. In general, an attempt to place the existing CDV literature within a more rigorous evolutionary framework is to be welcomed,  and helps to differentiate this article from a number of  comprehensive well written reviews on CDV which have been published in the past 3-4 years. The main weakness to this approach is the paucity of high quality data, which is available to review, especially with respect to mechanistic in vitro or in vivo studies or large-scale whole genome sequencing of wildlife samples. In general the study is well written but is often repetitive with major flaws in the use of references and presentation of data and more concise writing would also be appreciated.

Major comments

1.      Please completely reassess the use of references. There are many, many examples of references being used in appropriately in that they often appear to bear no relation to what is being discussed in the preceding sentence. By line 111, reference 1 which is itself a review (book chapter) has already been cited 12 times.  Just to state a couple of the most egregious examples: Line 353, reference 93; Lines 381-382, reference 20; line 412, reference 90.

R/. We agree to the reviewer and update some of the references according to the recommendation.

Line 66, 74: (Whelan, Barr, and Wertz 2004)

Line 77: se corrige formato de referencia (7)

Line 89: (Russell, Jardetzky, and Lamb 2001)

Line 98: (Bringolf et al. 2017)

Line 105: (Summers, Greisen, and Appel 1979)

Line 106: (Delpeut, Noyce, and Richardson 2014)

Line 110: (Koutinas et al. 2004)

Line 111: (Rudd, Cattaneo, and von Messling 2006)

Line 353:  (Tatsuo, Ono, and Yanagi 2001)

Line 381: We delete reference 20

Line 382 :  (Fen et al. 2016); (Qiu et al. 2011)(Cao et al. 2017)

Line 412: We delete reference 90

 2.      The introduction needs to be modifies as it currently reads like part of a PhD thesis i.e. before describing the properties of CDV, I would suggest to start with a small paragraph introducing the overall topic of the review. Then I would group the description of the proteins involved with the RNP and the M/F/H into two separate paragraphs instead of one paragraph per gene.

 R/. We agree to the reviewer. A short introductory paragraph was added (lines 51 -57)

 3.      Then I would group the description of the proteins involved with the RNP and the M/F/H into two separate paragraphs instead of one paragraph per gene.

R/. We accept the suggestion of the reviewer and modified the paragraph in two separate paragraphs. Lines 66-100.

 4.      For the section ‘Reservoirs vs Viral Adaptation: two different transmission sceneries’ (line 151), I would suggest that the authors start with the role of reservoirs in CDV epidemiology as the literature supporting this is much more extensive than the more speculative role of asymptomatic animals in CDV transmission.

We agree to the reviwer and modify the order of the paragraph:

Line 159:” The role of reservoirs in CDV epidemiology”

Line 198: “ Role of asymptomatic animals in CDV transmission”

Line 265: “Viral Adaptation: the role of specialist vs generalist strains”

 5.      For the section ‘CDV and Measles’, I would consider having a new title which is more self-explanatory and starting with a small introductory paragraph which introduces this subject.

R/. We agree to the reviewer and change the subtitle to: CDV and measles: Complementary infection model (line 353)

 A short introductory paragraph was added to this section (Lines 354-358).

 6.      It is not clear what the difference is between Tables 2 and 3 as some of the H amino acids listed in Table 3 also interact with CD150. I would suggest combining the references and alter the formatting so that the reference is given as a number to save on space

R/. We agree to the reviewer. We combined Tables 2 and 3. Line 457.

Also, the document reference to table was updated according to this change.

7.      I would consider deleting figure 3 as one of the weaknesses of including original data in a review is that the way the analysis was performed is not described, with no data for the reliability of the different branches or on what sequences were used for the analysis.

R/. We appreciate the reviewer suggestion regarding this Figure. We will explain that the Bayesian methodology throws the highest probability of the tree, therefore the frequency of each clade serves as branch support in terms of probabilities, so we do not talk about bootstrap and we added those frequencies in each clade (Figure 3).

A table with the sequences used for Figures 2 and 3 was added as a Supplementary material.

In addition, the position of Asia 3 looks to be incorrect and the dates seem to bear no relation to the historical literature.

R/. In relation to the Asia-3 lineage, there is a current discussion about its origin due to recombination processes between vaccine strains and circulating wild strains in Asia (Yuan et al., 2017), however when looking at the table of distances between these strains and the strains of the lineage North America -1 the aminoacidic divergence does not exceed 4% because of this in the tree they are grouped in the same clade. On the other hand, authors such as (Bhatt et al.2019) report the Asia-3 lineage as a well-defined clade, however, other authors  (Si et al 2010) and (Ke et al., 2015) had grouped these sequences in the Asia-2 lineage.

In relation to the dates presented in this tree, it should be clarified that the current presented molecular clock coincides with other previously reported analysis of CDV (Pomeroy et al., 2008), (Panzera et al., 2015), (Fischer et al., 2016). In the Bayesian analysis the most important thing is 95% higher probability density (HPD) interval, rather than the average showed in the graph. It is also important to keep in mind that this analysis  are based on hemagglutinin sequences available in the genbank, and not in the historical reports of the CDV such as that of Edward Jenner in the mid-1700s. A short Sentence to clarify this point has been included at the end of the paragraph (Line 524-527).

 Minor comments

1.      The title is too long and unwieldy, why not keep it to ‘Evolution and interspecies transmission of canine distemper virus’.

R/. We partially agree to the reviewer recommendation. However due to there are different CDV reviews author believe that the presented” title and Subtitle strategy” shows an important landmark of differentiation between our review and other previously published ones.

 2.      Please check the formatting of the references.

R/. References were included in ENDnote to assure the correct formatting according to the journal requirements.

3.      There are many English mistakes throughout the manuscript. Please do additional proofreading.

R/. An additional English proofreading to the entire manuscript was done

4.      Lines 71-72. The functions of all of the proteins has been described except the viral polymerase – maybe add that it replicates and transcribes the viral genome.

R/. We agree the reviewer. A short sentence were added. (Line 76)

 5.      Line 132. There is currently no evidence to suggest that the CDV strain responsible for the outbreak of distemper in lions in 1993/94 was adapted to non-canids. Please change

R/. Recent paper by Nikolin et al. 2017, shows that CDV from 1993 outbreak was indeed adapted to non-canids. The reference was added to support this sentence. (Lines 136-139)

 6.      Lines 147-148. This is a repetition of lines 134-136.

R/. We agree to the reviewer. The sentence was deleted. Lines 128-129

 7.      Lines 230-232. How does the proposal of CDV spread from USA to Europe during 1858 to 1913 match up with the extensive documentation of CDV outbreaks in Europe from 1760 e.g. see the famous description of CDV by Edward Jenner in 1809 (20895116)

 R/.  We agree to the reviewer. tMRCA results are based on sequences of hemagglutinin that have been recovered from clinical samples or isolates in cell culture, therefore these phylogenetic approaches are really genetic inference in which it is applied evolutionary models (algorithms) that can explain the most recent common ancestor, however, as discussed (Panzera et al., 2015), it can not be assumed that the origin of the CDV was at this time and in this region since it has an obligated selection bias. In the case of CDV, the most ancestral sequences belong to the vaccine strains isolated in the late 1950s.  

A short paragraph was added to the paper in order to clarify this bias in the analysis  (lines 524-527)

 8.      Lines 271-271. All of the available evidence shows that CDV has specialized in the order Caniformia, with infections of large felids largely representative of spill over events from infected canid species

R/. Although it is clear that CDV has specialized in the Caniformia order, it is important to continue studying if the antagonistic pleitropism is explaining the adaptations to the big cats as discussed by (Nikolin et al.2017). It is clear that CDV can be a succesful infection in other species and ecosystems and different strains of CDV can be stably adapted to these new orders by acquiring and fixing mutations. However as we stated, other  important features such as immune evasion in each host should be studied in a deep way, and in each affected host (Line 280-282).

 9.      Lines 366-371. This is written as if a third CDV receptor has already been discovered. Before ‘this new receptor should be further characterized’, it first needs to be identified. It should also be mentioned that in CDV-infected brains, neurons are more commonly infected than glial cells

R/. we agree to the reviewer and apologize for the ambiguous text. The text was modified in order to clarify this item. (Lines 366-371).

 10.  Line 412. This reviewer is unaware of any countries in which MV vaccination has been discontinued due to successful eradication. Indeed, it is worth noting that there is yet no official eradication program, although on a regional level e.g. the Americas, endemic MV transmission has been stopped

 R/. We agree to the reviewer. There was a mistake in redaction of the idea. The paragraph was modified. Lines 420-423

    11. Figure 2. Please re-check the position of the Asia 3 lineage as this does look to be correct.

R/. As previously discussed in figure 3, In relation to the Asia-3 lineage, there is a current discussion about its origin due to recombination processes between vaccine strains and circulating wild strains in Asia (Yuan et al., 2017). However when looking at the table of distances between these strains and the strains of the lineage North America -1 the aminoacidic divergence does not exceed 4% because of this in the tree they are grouped in the same clade. On the other hand, authors such as (Bhatt et al.2019) report the Asia-3 lineage as a well-defined clade, however, other authors  (Si et al 2010) and (Ke et al., 2015) had grouped these sequences in the Asia-2 lineage.

Reviewer 2 Report

Valencia et al - Viruses

 The aim of this article is to review the interspecies transmission of canine distemper virus (CDV), a virus that is known to infect diverse species. Although the review has 127 references, relatively few of them relate to the proposed focus of the review (cross-species transmission).  It fails to address the ecology of the virus, why it is able to jump species, evading host innate and adaptive immunity. What role does the V protein play? What about F protein fusogenicity? The review focuses almost entirely on studies looking at H in which mutations in H which have been recorded in a number of non-canid species are stated as being the single cause of cross-species transmission and pathogenicity (with very limited supporting data).  Overall the review is a missed opportunity.

 A general issue with the review is the tendency to either cite other reviews/text books, or use inappropriate citations. In several places the authors simply cite “1. MacLachlan N, Dubovi E, Fenner F. Paramyxoviridae2011. 299-325 p.” This not a true source, it is a chapter that already cites the true source studies. See for example lines 105-111 where the authors simply cite this book chapter repeatedly. In several places, citations are made but when the citations are followed, the cited papers do not actually support the statements made in the review. It would be unhelpful for a review to be published that has a tendency for flawed interpretation or over-interpretation of the published literature. I would expect a higher standard of critical analysis of the literature in  a review for Viruses.

 Following is a line by line commentary on the article:

 Line 49 - The authors begin with a general introduction to CDV, describing the genetic make-up of the virus and the viral proteins encoded. I would question whether this is actually necessary for a review of CDV interspecies transmission, it is covered at length in many other articles. It would be more useful to begin with an overview of the ecology of CDV. It would be helpful if the authors were more precise in the introduction of the virus, explaining to the reader that the species name has been amended to “canine morbillivirus”, just as the species name for measles virus is now “measles morbillivirus”. Measles virus is generally abbreviated to MeV, seal distemper is actually “phocine” distemper virus (PDV), peste des petits ruminants (PPRV) and rinderpest virus (RPV). Note the lower case “p” in peste.

 Line 78 - the authors should define the units of substitution rate (nucleotide substitutions per site, per year).

 Line 80 – define “sentinel” cells, lower case “n” for nectin-4

 Line 81 – “After CDV infection, neutralizing antibodies against the protein H confer immunity for life [11]” – Not only has this not been shown, the reference cited makes no reference to neutralising antibodies against H and long-term immunity. It is an assumption.

 Line 81 – define the percentage figures for divergence. Identity? Similarity?

 Line – 96 “synthesised” not “synthetized”

 Line 103 – “The main infection route is through the aerosolization of nasal secretions and body excretions such as urine and feces [14].” This is clumsy wording.  As transmission is seldom witnessed, it would be fairer to say that transmission is “likely” or “thought to be” through contact with fomites or via the aerosol route. This should be backed up by references that investigate transmission routes.

 Line 132 – “Most notably, a strain adapted to non-canids caused a fatal epidemic” – This is imprecise, the authors did not demonstrate that the strain grew more efficiently or was more pathogenic in non-canids. At best, they showed that there were nucleotide changes in the genomes of the non-canid viruses that were consistent with expansion of selected variants in non-canid hosts, an important distinction. 

 Line 149 – “by infecting wildlife in danger of conservation” this does not make sense

 Line 151 – “sceneries” is the plural of scenery and seems out of place. Do you mean scenarios?

 Line 163 – again, “panoramas” seems out of place

 Line 158 -  “…these individual act as CDV reservoirs [42, 43]. Recently…”. I would suggest “potential” reservoirs for CDV…

 Line 169 – spurious extra “individuals”

 Line 179 – whole paragraph beginning “'Silent' CDV epidemics have been observed in the Serengeti-Mara ecosystem, where serological analyses ….”.  Can the spread of a virus that does not cause disease actually be defined as an “epidemic”? Please also note that hyenas and lions are not humans and hence do not have “symptoms”, only humans have “symptoms”, they experience and articulate the symptom. Animals display “clinical signs” of disease.

 Line 196 -  “….and lion when infected with CDV canid strains [25].” The authors of this study never demonstrated that the lions were infected with a canid strain of CDV, rather they looked at associations between a limited number of sequences (13) and suggested that the virus that was present in non-canids was distinct from the virus in canids.

 Line 209 -  “..the ability of the wild strains to re-infect or reactivate the infection in the presence of neutralizing antibodies that cannot neutralize the virus, caused by mutations that change surface antigens - [30].” In this paragraph on CDV vaccines, the authors cite a reference as a source for the activities of neutralising antibodies. The authors of the study did not look at neutralising antibodies, only phylogeny. Moreover, in the paragraph, they fail to cover other CDV vaccines, including the widely used canarypox vectored vaccines for endangered species that may be susceptible to live attenuated strains of virus.

 Line 221 -  “CDV spreading from wildlife to domestic canids is also possible [68].” Again the authors are citing a review as a source paper, please correctly identify and acknowledge the source.

 Line 248 – scenario?

 Line 262 – Please revise the punctuation of this sentence.

 Line 271 -  “CDV is thought to have specialized in two major host orders: Caniformia and Feliformia, in which the virus quasispecies with better host fitness are considered specialized strains.” The authors should provide evidence in support of this statement.

 Line 388 -  “…and in the third passage author cannot find mutations in H but…” Please revise.

 Line 405 – Paragraph is factually incorrect. “It is well known that experimental MV vaccination in pups usually leads to an immune response with cross-reactive activity to CDV even in the presence of maternal antibodies [104].” The reference cited is an experiment in mice looking at CTLs, do the authors mean mouse pups or dogs? If mice, then the reference does not look at neutralising antibodies, if dogs, then the reference is incorrect. Historically, MeV vaccines have been used in dogs to circumvent maternally derived antibodies against CDV, but the study the authors cited has nothing to do with dogs.

 Line 420 -  “Studies using the H  gene show a worldwide distribution of CDV genetic”. Perhaps the authors mean “genotypes”.

 Line 415 and Figure 2 – What is the point of the section on genetic variability of CDV and the phylogenetic tree in regard to cross-species transmission of CDV? The authors conflate two distinct topics, 1) genetic variability and 2) genetic changes associated with host adaptation. They do not assess whether there is a relationship between the degree of genetic variability and likelihood of cross species transmission. 

 Lines 432 and 433 – Please define the percentages.

 Line 520 – “the lack epidemiological in certain population”. This makes no sense.

 Line 528 -  “…positions (519I and 549H) was only observed in non-canid strains and was related to high mortality in these species.” The authors did not show that the mutations in H were the cause of high mortality.

 Author Response

RESPONSE TO REVIEWERS COMMENTS (Manuscript ID: viruses-482333)

Evolution and interspecies transmission of Canine Distemper Virus – an outlook of the diverse evolutionary landscapes of a multi-host virus

July Duque-Valencia, Nicolás Sarute, Ximena A. Olarte-Castillo, Julián Ruíz-Sáenz*

 Dear Editor

Please find below our point by point responses to the comments regarding our Manuscript ID: viruses-482333, formerly entitled “Evolution and interspecies transmission of Canine Distemper Virus – an outlook of the diverse evolutionary landscapes of a multi-host virus”. The changes are highlighted in Yellow in the file.

We would like to thank the Reviewer for their helpful suggestions, for critical analysis of the manuscript, and for providing new discussion topics.

REVIEWER 2

·         The aim of this article is to review the interspecies transmission of canine distemper virus (CDV), a virus that is known to infect diverse species.

R/. Although it is well know the interspecies transmission of CDV, there is not consolidated knowledge about the main genomic and epidemiological drivers of this “jump”. The paper focuses mainly in the possibilities that allow the interspecies transmission by discussing diverse results from different molecular and epidemiological case scenarios in which the CDV has jump between species by highlighting the role of 1). Molecular adaptation, 2) the unidentified role of viral reservoirs in the wild and 3). the role of asymptomatic animals in domestic and wild populations. Besides, In order to understand the role of viral adaptation, we discuss the role of the full genome analysis as key to understand mutational hotspot in CDV genome in addition to the single gene sequencing of the viral hemagglutinin. A short introductory paragraph was added (Lines 54-57)

 ·         Although the review has 127 references, relatively few of them relate to the proposed focus of the review (cross-species transmission).

R/. We agree the reviewer that there are relatively few references about the main topic of the review. In fact, this is the first review that try to propose and support the different drivers that allow interspecies transmission. Due to the lack of specific experimental information, we decide to analyze and integrate the different isolated information, from clinic cases, outbreaks in different wild and captive species and its relation to the well know domestic epidemiology. In fact, although it was well known the role of the CDV in the Serengeti ecosystems outbreak in late 90’s, only recently was published the molecular approximation to the evolution of the CDV in such ecosystem (Nikolin et al., 2016).

 ·         It fails to address the ecology of the virus, why it is able to jump species, evading host innate and adaptive immunity.

R/. We partially agree to the reviewer. One of the main conclusions of the review is that interspecies transmission of the CDV in wild ecosystems and Between wildlife and domestic animals, are more complex than previously reported in individual analysis. The main conclusion highlights the critical necessity of interdisciplinary research in wild and domestic ecosystems to allow a better understanding of the ecology of the CDV. To our knowledge, there are very few papers describing this focus that goes from the epidemiolocal data, to the molecular analysis in full genome.

The role of the adaptative immunity and host immune response to the CDV, has been widely discussed and studied in the domestic dog. However due to the impossibility to experimentally study this topic in wildlife, it is only possible to speculate how the evasion of the host immune system is involve in CDV infection in different species and its role in the pathogenesis. We included a short paragraph highlighting this gap in the knowledge (Lines 154-157)

 ·         What role does the V protein play? What about F protein fusogenicity? The review focuses almost entirely on studies looking at H in which mutations in H which have been recorded in a number of non-canid species are stated as being the single cause of cross-species transmission and pathogenicity (with very limited supporting data). 

R/. we partially agree to the reviewer. We could not find enough information about the role that CDV V and/or F proteins plays in the interspecies transmission, pathogenesis among different species. There are a lack of information about the study of the full CDV genome and its adaptation to different species. Only two papers try to explain the genome adaptation in  human cells or in wild fauna (Otsuki, Nakatsu, et al. 2013; Otsuki, Sekizuka, et al. 2013) /(Nikolin et al. 2016). A short sentence was added to the paragraph for a better understanding of this possible bias – (Lines 491-494).

In the other hand, CDV full genome has not been widely studied. There are only 103 full CDV genomes included the NCBI database and only 10% of this are from Wildlife. Thats why we propose the importance of performing CDV full genome analysis in domestic and wild animals (Lines 491-494).

 ·         Overall the review is a missed opportunity.

R/. We disagree to this affirmation. It is clear that CDV interspecies transmission is a intricate issue. Due to the lack of field information on the eco-epidemiology of the interspecies transmission of CDV, our paper focused in the Evolutive differences that emphasize the relevance of CDV full genome studies as in wild life as in domestic dogs. The authors has different Academic background (Phylogeneticist, Veterinarians, Molecular Epidemiologist, Virologist) to generate an interdisciplinary review that could integrate the existent knowledge in CDV interspecies transmission as we propose in the conclusion (Lines 585-586)

 ·         A general issue with the review is the tendency to either cite other reviews/text books, or use inappropriate citations. In several places the authors simply cite “1. MacLachlan N, Dubovi E, Fenner F. Paramyxoviridae2011. 299-325 p.” This not a true source, it is a chapter that already cites the true source studies. See for example lines 105-111 where the authors simply cite this book chapter repeatedly.

R/. We agree to the reviewer and apologize for this mistake. We change some references to be most accurate in referencing

Line 66, 74: (Whelan, Barr, and Wertz 2004)

Line 77: We corrected reference

Line 89: (Russell, Jardetzky, and Lamb 2001)

Line 98: (Bringolf et al. 2017)

Line 105: (Summers, Greisen, and Appel 1979)

Line 106: (Delpeut, Noyce, and Richardson 2014)

Line 110: (Koutinas et al. 2004)

Line 111: (Rudd, Cattaneo, and von Messling 2006)

 ·         In several places, citations are made but when the citations are followed, the cited papers do not actually support the statements made in the review. It would be unhelpful for a review to be published that has a tendency for flawed interpretation or over-interpretation of the published literature. I would expect a higher standard of critical analysis of the literature in  a review for Viruses.

R/. We agree to the reviewer and apologize for this mistake. We corrected some references.

Line 353:  (Tatsuo, Ono, and Yanagi 2001)

Line 381 Reference 20 was erased.

Line 382 :  (Fen et al. 2016); (Qiu et al. 2011)(Cao et al. 2017)

Line 412: Reference 20 was deleted.

 ·         Line 49 - The authors begin with a general introduction to CDV, describing the genetic make-up of the virus and the viral proteins encoded. I would question whether this is actually necessary for a review of CDV interspecies transmission, it is covered at length in many other articles.

R/. We believe that the introduction should describe the functions of proteins because it supports the conclusion of the article to work with complete genomes and not just with hemagglutinin. We also group the description of the genes into two paragraphs (Lines 66-100).

 ·         It would be more useful to begin with an overview of the ecology of CDV.

R/. We agree to the reviewer, and added a short introductory paragraph to the paper (Lines 53-57).

 ·         It would be helpful if the authors were more precise in the introduction of the virus, explaining to the reader that the species name has been amended to “canine morbillivirus”, just as the species name for measles virus is now “measles morbillivirus”. Measles virus is generally abbreviated to MeV, seal distemper is actually “phocine” distemper virus (PDV), peste des petits ruminants (PPRV) and rinderpest virus (RPV). Note the lower case “p” in peste.

R/. We agree to the reviewer and modify the paragraph including the current taxonomy names according to the ICTV 2018 release (Lines 50-53).

 ·         Line 78 - the authors should define the units of substitution rate (nucleotide substitutions per site, per year).

R/. We agree to the reviewer and modify the paragraph (Line 82-83).

The hemagglutinin gene has 1824 nucleotides and encodes for the protein H, which shows the highest variability in CDV when compared to other morbilliviruses [1]. This gene has the highest divergence within CDV genome with substitution rates between 5.4 x10-4–1.8x10-3 nucleotide substitutions per site, per year [1].

 ·         Line 80 – define “sentinel” cells, lower case “n” for nectin-4

R/. We agree to the reviewer and modify the paragraph. The word “Sentinel was erased and the lower case was checker for nectin-4 in all document. (Line 84-85)

The protein H determines the viral tropism and initiates the infection  by binding to the signaling lymphocytic activation molecule (SLAM) receptor in immune cells [2,3], and nectin-4 receptor in epithelial cells [4].

 ·         Line 81 – “After CDV infection, neutralizing antibodies against the protein H confer immunity for life [11]” – Not only has this not been shown, the reference cited makes no reference to neutralising antibodies against H and long-term immunity. It is an assumption

R/. We partially agree to the reviewer. The term “Immunity for life” could be ambiguous. By evaluating the longest period of time after initial infection with modified live virus , it has been shown the persistence of antibody in dogs until 14 years (Proceedings of the Second Merial European Comparative Vaccinology Symposium : 'Vaccination Challenges in Ageing Populations'). That’s why it has been assumed that when dogs recover from natural infection/disease due to CDV, they develop lifelong immunity to these diseases. We modify the sentence and change the reference (Line 85-87).

 ·         Line 81 – define the percentage figures for divergence. Identity? Similarity?

R/. We believe that it is more appropriate to speak of divergences when comparing wild strains with vaccine strains in phylogenetic analyzes, however, we will add the percentage of identity (Line 88-89)

 ·         Line – 96 “synthesised” not “synthetized”

R/. We agree and modify. (Line 98). A full edition of the paper was done.

 ·         Line 103 – “The main infection route is through the aerosolization of nasal secretions and body excretions such as urine and feces [14].” This is clumsy wording.  As transmission is seldom witnessed, it would be fairer to say that transmission is “likely” or “thought to be” through contact with fomites or via the aerosol route. This should be backed up by references that investigate transmission routes.

R/. We agree the reviewer and modify the paragraph (Lines 104-106)

The main route of efficient infection is thought to be through contact with fomites or via the aerosol route through nasal secretions [5]; however transmission through nasal contact with bodily excretions such as urine and feces is also likely [6].

 ·         Line 132 – “Most notably, a strain adapted to non-canids caused a fatal epidemic” – This is imprecise, the authors did not demonstrate that the strain grew more efficiently or was more pathogenic in non-canids. At best, they showed that there [7], an important distinction.

R/. We agree to the reviewer and apologize for the imprecision. The paragraph was corrected for a better presentation of the information (Lines 135-139).

However the main conclusion of Nikolin et al 2017 paper stated textually: “Our findings are consistent with an epidemic in lions and hyaenas caused by CDV variants better adapted to non- canids than canids, but not with the recent spillover of a dog strain. Our study reveals a greater complexity of CDV molecular epidemiology in multihost environments than previously thought.”

 ·         Line 149 – “by infecting wildlife in danger of conservation” this does not make sense

R/: We agree to the reviewer and modify the sentence. (Lines 152-153)

Those data allow us to propose that CDV, by infecting endangered wildlife species, decreases the populations of these species, facilitating its extinction..

 ·         Line 151 – “sceneries” is the plural of scenery and seems out of place. Do you mean scenarios?

R/. We agree to the reviewer. All document was checked and corrected.

 ·         Line 163 – again, “panoramas” seems out of place

R/. We agree to the reviewer. All document was checked and corrected.

 ·         Line 158 -  “…these individual act as CDV reservoirs [42, 43]. Recently…”. I would suggest “potential” reservoirs for CDV…

R/. We agree to the reviewer. The word “Potential was added”. (Line 204)

 ·         Line 169 – spurious extra “individuals”

R/. We agree to the reviewer. The spurious extra “individuals” were deleted. (Line 216)

 ·         Line 179 – whole paragraph beginning “'Silent' CDV epidemics have been observed in the Serengeti-Mara ecosystem, where serological analyses ….”.  Can the spread of a virus that does not cause disease actually be defined as an “epidemic”?

R/. 'Silent' CDV epidemics has been stated as a phenomenon in which serological analyzes show periods in which exposure to CDV increased (Harrison et al. 2004; Munson et al. 2008; Viana et al. 2015). References were included in the paragraph (Line  226-229)

 ·         Please also note that hyenas and lions are not humans and hence do not have “symptoms”, only humans have “symptoms”, they experience and articulate the symptom. Animals display “clinical signs” of disease

R/. We agree to the reviewer. We replace the Word symptom in all document by “clinical signs”

 ·         Line 196 -  “….and lion when infected with CDV canid strains [25].” The authors of this study never demonstrated that the lions were infected with a canid strain of CDV,

R/. We agree to the reviewer and modify the paragraph according to the reference (Nikolin et al.2016). Line 243.

 ·         Rather they looked at associations between a limited number of sequences (13) and suggested that the virus that was present in non-canids was distinct from the virus in canids

R/. We agree to the reviewer and understand the possible bias. However, Although the number of sequences is small, it is the only paper that analyzed complete genomes in wildlife from this evolutionary approach. The sentence was modified to state that this is a “possibility” not a totally stablished conclusion (Line 243-244)

 ·         Line 209 -  “..the ability of the wild strains to re-infect or reactivate the infection in the presence of neutralizing antibodies that cannot neutralize the virus, caused by mutations that change surface antigens - [30, ].” In this paragraph on CDV vaccines, the authors cite a reference as a source for the activities of neutralising antibodies. The authors of the study did not look at neutralising antibodies, only phylogeny.

R/. We agree to the reviewer. However, researchers working in CDV phylogeny have the hypothesis that wild strains are infecting animals with vaccination schemes as has been reported in cases of wildlife (Chinnadurai et al. 2017) and domestic dogs (Li et alt.2014) and this is due to escape mutants or to the fact that the amino acid divergence in hemagglutinin is so high that as a consequence there are structural modifications in the protein that cause the antibodies generated by the Vaccine strains do not neutralize wild strains (Iwatsuki et al.2000). The reference were modified by other that shows neutralization of vaccinal antibodies against wild strains. (Iwatsuki et al.2000, Anis et al.2018) Line 256.

 ·         Moreover, in the paragraph, they fail to cover other CDV vaccines, including the widely used canarypox vectored vaccines for endangered species that may be susceptible to live attenuated strains of virus.

R/. We agree to the reviewer, modify the sentence and include a new reference of the variability of immune response by the Canarypox vectored vaccine in captive Pantera tigris (Sadler et al., 2016). (Lines 258-259).

It is important to state that although Canarypox vectored vaccines could be safer that the live attenuated vaccines (MLV) of  CDV, due to its H gene genetic composition (Almost all are based on the Onderstepoort strain – America-1 Lineage - Patent US7507416) it can be reasonable to suppose the same “low neutralization” event that occurs as in dogs on which the MLV vaccine produce high titers of antibodies and does not neutralize the wild type field strains (Riley and Wilkes. 2015, Anis et al.2018).

 ·         Line 221 -  “CDV spreading from wildlife to domestic canids is also possible [68].” Again the authors are citing a review as a source paper, please correctly identify and acknowledge the source.

R/. We agree to the reviewer and include new references (Kapil et al.2008, Kapil and Yeary 2011) Line 165.

 ·         Line 248 – scenario?

R/. We modify the sentence. (Line 187)

 ·         Line 262 – Please revise the punctuation of this sentence

Concepts such as fitness, defined as ‘the capacity of a virus to produce infectious progeny in a given environment’ [8], quasispecies ‘collections of closely related viral genomes subjected to a continuous process of genetic variation, competition and selection’ [9], coevolution ‘evolutionary change in a trait of the individuals in one population in response to a trait of the individuals of a second population, followed by an evolutionary response by the second population to the change in the first’ [10], and antagonistic pleiotropy ‘a beneficial mutation in one environment is either harmful or neutral in another environment, or mutations that are neutral in the environment in which they arose are deleterious in another’ [11] have been shown to play a role on the adaptation and evolution of CDV in its different hosts.

 R/. The punctuation of the paragraph was improved. (Lines 271-279)

 ·         Line 271 -  “CDV is thought to have specialized in two major host orders: Caniformia and Feliformia, in which the virus quasispecies with better host fitness are considered specialized strains.” The authors should provide evidence in support of this statement.

R/. We agree to the reviewer. The sentence needed references. Thos were included at the end of the line (Nikolin et al. 2012, Nikolin et al.2016). Lines 281-282

 ·         Line 388 -  “…and in the third passage author cannot find mutations in H but…” Please revise

R/. The sentences were modified and reference where added (Otsuki et al. 2013) (Lines 396-397)

 ·         Line 405 – Paragraph is factually incorrect. “It is well known that experimental MV vaccination in pups usually leads to an immune response with cross-reactive activity to CDV even in the presence of maternal antibodies (Appel et al.1984, de Vries et al.1988) .” The reference cited is an experiment in mice looking at CTLs, do the authors mean mouse pups or dogs? If mice, then the reference does not look at neutralising antibodies, if dogs, then the reference is incorrect. Historically, MeV vaccines have been used in dogs to circumvent maternally derived antibodies against CDV, but the study the authors cited has nothing to do with dogs

R/. We agree to the reviewer and apologize for this mistake. A set of original references about the use of MV Vaccine in puppies were added (Baker et al., 1966, Brown et al., 1972, Appel et al., 1984, de Vries et al., 1988). Lines 413-414.

 ·         Line 420 -  “Studies using the H  gene show a worldwide distribution of CDV genetic”. Perhaps the authors mean “genotypes”.

R/. We agree to the reviewer. The sentence has an involuntary mistake (Line 429)

 ·         Line 415 and Figure 2 – What is the point of the section on genetic variability of CDV and the phylogenetic tree in regard to cross-species transmission of CDV? The authors conflate two distinct topics, 1) genetic variability and 2) genetic changes associated with host adaptation. They do not assess whether there is a relationship between the degree of genetic variability and likelihood of cross species transmission.

R/. Although the phylogenetic classification in “Lineages” is given by the genetic variability only in H gene, it is clear that different hosts could be infected by the same lineage, reinforcing the evidence that epistatic interactions and/ or compensatory changes in the whole genome could be associated with species barrier jump and pathogenicity (von Messling et al.2003, Ke et al. Ke et al.2015, Fisher et al.2015). In addition, sites under positive selection could help to explain the species barrier jump and pathogenicity (Mccarthy et al., 2007).

 ·         Lines 432 and 433 – Please define the percentages

R/. The percentages for each sublineage are not yet defined, so the subclades with a bootstrap of 70% are established in the phylogenetic tree. The sublineage is assigned with the strains that have an amino acid identity of 98%, This methodology has been recently adapted for CDV and still there are not enough studies that can support us exact percentages as they are for measles. Lines 441-443

 ·         Line 520 – “the lack epidemiological in certain population”. This makes no sense.

R/. We agree to the reviewer and modify the sentence.  Line 533

 ·         Line 528 -  “…positions (519I and 549H) was only observed in non-canid strains and was related to high mortality in these species.” The authors did not show that the mutations in H were the cause of high mortality

R/. We agree to the reviewer and modify the sentence according to the original paper (Line 540-542)

…”Our finding that strains encoding 519I/549H cause fatal outcomes only in noncanids (lion and hyaena) during the 1993/1994 epidemic but not in canids is consistent with previous reports of the virulence of this strain type to noncanids”

 Reviewer 3 Report

This review paper describes the important characteristics of CDV that make the virus great threat to many wild animal species. It also explains importance of H protein antigenicity that determines adaptation of the virus to different hosts and H and V proteins in pathogenicity of the virus. The paper contains important information and useful in the field of CDV research for protection of both wild and domestic animal. Therefore, I would recommend this paper for publication in the journal after some minor corrections.

Comments;

In the section "CDV and measles", it is not clear why it is necessary to explain measles for the aim of this paper (explanation of evolution and interspecies transmission of CDV). It would be helpful for readers if the authors add clear explanation at the beginning of the section, as to how description of measles virus biology would help describing evolution and adaptation of CDV among different species. 

Specific comments;

Line 30: correct "drive" to "driven"

Line 55: "six genes" should be corrected to "six gene regions containing 8 genes" since P gene region also contains C and V genes

 Author Response

RESPONSE TO REVIEWERS COMMENTS (Manuscript ID: viruses-482333)

Evolution and interspecies transmission of Canine Distemper Virus – an outlook of the diverse evolutionary landscapes of a multi-host virus

July Duque-Valencia, Nicolás Sarute, Ximena A. Olarte-Castillo, Julián Ruíz-Sáenz*

 Dear Editor

Please find below our point by point responses to the comments regarding our Manuscript ID: viruses-482333, formerly entitled “Evolution and interspecies transmission of Canine Distemper Virus – an outlook of the diverse evolutionary landscapes of a multi-host virus”. The changes are highlighted in Yellow in the file.

We would like to thank the Reviewer for their helpful suggestions, for critical analysis of the manuscript, and for providing new discussion topics.

REVIEWER 3

This review paper describes the important characteristics of CDV that make the virus great threat to many wild animal species. It also explains importance of H protein antigenicity that determines adaptation of the virus to different hosts and H and V proteins in pathogenicity of the virus. The paper contains important information and useful in the field of CDV research for protection of both wild and domestic animal. Therefore, I would recommend this paper for publication in the journal after some minor corrections.

R/. Thank you very much for your appreciation on our paper.

 Comments:

In the section "CDV and measles", it is not clear why it is necessary to explain measles for the aim of this paper (explanation of evolution and interspecies transmission of CDV). It would be helpful for readers if the authors add clear explanation at the beginning of the section, as to how description of measles virus biology would help describing evolution and adaptation of CDV among different species.

R/. We agree to the reviewer. The title of this section was improved (Line 353) and a short introductory paragraph was added (Lines 354-358)

 Specific comments;

Line 30: correct "drive" to "driven"

R/. We agree to the reviewer and Correct the word.

 Line 55: "six genes" should be corrected to "six gene regions containing 8 genes" since P gene region also contains C and V genes

R/. We agree to the reviewer. The paragraph was improved according to the recommendation (Line 61-62)

Round  2

Reviewer 1 Report

n/a

Author Response

RESPONSE TO REVIEWERS COMMENTS (Manuscript ID: viruses-482333)

Evolution and interspecies transmission of canine distemper virus – an outlook of the diverse evolutionary landscapes of a multi-host virus

July Duque-Valencia, Nicolás Sarute, Ximena A. Olarte-Castillo, Julián Ruíz-Sáenz*

Dear Editor

Please find below our point by point responses to the comments regarding our Manuscript ID: viruses-482333, formerly entitled “Evolution and interspecies transmission of canine distemper virus – an outlook of the diverse evolutionary landscapes of a multi-host virus”. The changes are highlighted in Yellow in the file.

We would like to thank the Reviewer for their helpful suggestions, for critical analysis of the manuscript, and for providing new discussion topics.

REVIEWER 1.

R/. All changes and suggestion included in attached PDF file has been take into account. Thanks for the amazing Review!

Reviewer 2 Report

Please re-check the manuscript for inappropriate use of upper case letters (eg in the title), hyphens, MeV (not MV) etc. Foxes don't have symptoms....

Author Response

RESPONSE TO REVIEWERS COMMENTS (Manuscript ID: viruses-482333)

Evolution and interspecies transmission of canine distemper virus – an outlook of the diverse evolutionary landscapes of a multi-host virus

July Duque-Valencia, Nicolás Sarute, Ximena A. Olarte-Castillo, Julián Ruíz-Sáenz*

 Dear Editor

Please find below our point by point responses to the comments regarding our Manuscript ID: viruses-482333, formerly entitled “Evolution and interspecies transmission of canine distemper virus – an outlook of the diverse evolutionary landscapes of a multi-host virus”. The changes are highlighted in Yellow in the file.

We would like to thank the Reviewer for their helpful suggestions, for critical analysis of the manuscript, and for providing new discussion topics.

REVIEWER 2

·         Please re-check the manuscript for inappropriate use of upper case letters (eg in the title), hyphens, MeV (not MV) etc. Foxes don't have symptoms....

R/. We have revised the entire document. Thanks for the amazing Review!